# Supervised breast cancer prediction using integrated dimensionality reduction convolutional neural network

**HuanQing Xu[1], Xian Shao[2,3], Shiji Hui[4], Li Jin[1] ***

**1** The School of Medical Information Engineering, Anhui University of Chinese Medicine, Hefei, China,
**2** NHC Key Laboratory of Hormones and Development, Chu Hsien-I Memorial Hospital and Tianjin Institute of Endocrinology, Tianjin Medical University, Tianjin, China, **3** Tianjin Key Laboratory of Metabolic Diseases, Tianjin Medical University, Tianjin, China, **4** School of Medical Information Engineering, Guangzhou University of Chinese Medicine, Guangzhou, Guangdong, China

* jinli19741223@163.com

**Data Availability Statement:** The data are available from the website (http://www.andrewjanowczyk.com/use-case-6-invasive-ductal-carcinoma-idc-segmentation/).

## Abstract

### Objectives

Breast cancer is a major health problem with high mortality rates. Early detection of breast caner will promote treatment. A technology that determines whether a tumor is benign desirable. This article introduces a new method in which deep learning is used to classify breast cancer.

### Methods

A new computer-aided detection (CAD) system is presented to classify benign and malignant masses in breast tumor cell samples. In the CAD system, (1) for the pathological data of unbalanced tumors, the training results are biased towards the side with the larger number of samples. This paper uses a Conditional Deep Convolution Generative Adversarial Network (CDCGAN) method to generate small samples by orientation data set to solve the imbalance problem of collected data. (2) For the high-dimensional data redundancy problem, this paper proposes an integrated dimension reduction convolutional neural network (IDRCNN) model, which solves the high-dimensional data dimension reduction problem of breast cancer and extracts effective features. The subsequent classifier found that by using the IDRCNN model proposed in this paper, the accuracy of the model was improved.

### Results

Experimental results show that IDRCNN combined with the model of CDCGAN model has superior classification performance than existing methods, as revealed by sensitivity, area under the curve (AUC), ROC curve and accuracy, recall, sensitivity, specificity, precision, PPV,NPV and f-values analysis.

### Conclusion

This paper proposes a Conditional Deep Convolution Generative Adversarial Network (CDCGAN) which can solve the imbalance problem of manually collected data by

**Funding:** The research is supported by the Provincial Natural Science Research Key Project of Anhui Universities (No. KJ2020A0443), the Key Project of Humanities and Social Sciences Research in Anhui Universities (No.SK2021A0329), the Outstanding Young Backbone Talents of Anhui University Visiting and Training Abroad Project (No. gxgwfx2019026), the Provincial Teaching Research Key Project of Anhui Province (No. 2020jyxm1018), the Natural Science Key Project of Anhui University of Chinese Medicine (No. 2020zrzd17), and the Quality Engineering Project of Anhui Province (No. 2021xsxxkc140).

**Competing interests:** The authors declare that they have no known competing financial interests or personal relationships that could have appeared to influence the work reported in this paper.

directionally generating small sample data sets. And an integrated dimension reduction convolutional neural network (IDRCNN) model, which solves the high-dimensional data dimension reduction problem of breast cancer and extracts effective features.

## Introduction

Breast cancer is one of the most common cancers worldwide [1]. With the development of artificial intelligence, neural networks have featured prominently in image recognition and have potential pioneering applications in diagnosis of breast cancer [2]. Characterization and evaluation of visual images introduces significant challenges to the doctor's evaluation. In the intelligent medical diagnosis system, experts not only have confidence in features, but also in visual images, which improves diagnoses' quality [3]. Convolutional neural network (CNN) is the most widely used neural network in image recognition. This artificial neural network possesses multiple hidden layers and is based on research on cells of the cat's visual cortex. CNN is comprised of a convolutional layer, pooling layer, fully connected layer, and activation function. CNN achieves dimensionality reduction by designing convolution kernels and pooling kernels, so they can process complex data with limited resources. Because of the invariance of translation, rotation, scaling and other forms, it is widely used in image recognition. Neural network models with excellent performance require excellent training datasets. But in real life, neural networks are limited by sample imbalance, especially in tumor sections, where imbalance between samples is very high. This problem is traditionally solved by using positive sample oversampling and negative sample under sampling. However, this results in loss of sample information. Yun Jiang [4] et al designed a novel CNN that includes a convolutional layer, a small SE-ResNet module, and a fully connected layer, achieving similar performance with fewer parameters. Alom [5] et al. proposed a method of classifying breast cancer using the initial residual convolutional neural network (IRRCNN) model. IRRCNN is a powerful deep Convolutional Neural Network (DCNN) model that combines the advantages of inception network (Inception-v4), residual network (ResNet) and recursive convolutional neural network (RCNN). IRRCNN has superior performance relative to equivalent inception networks, residual networks, and RCNNs used in object recognition.

There are many studies on artificial intelligence recognition of breast cancer. An evaluation of 3 strong novel deep learning-based approaches for automated examination of tumor cellularity in H/E stained, post-treatment breast surgical specimens found that deep learning-based methods can reduce burden on pathologists, enhance diagnoses, and improve breast cancer outcomes [6]. In addition, A technique developed by Abdelhameed Ibrahim [7] uses thermal imaging technology for non-invasive, non-contact cancer screening and can detect tumors early by measuring temperature distribution on the two breasts. An advantage of this approach is its invasive and non-contact nature. However, it involves a lot of data preprocessing to generate training features. Furthermore, Yasmeen Mourice George [8] developed a highly accurate breast cancer recognition machine learning system based on neural network (NN) and support vector machine (SVM). A system developed by MinhThanh Vo [9] first encodes global features extracted from the directional gradient histogram in a multilayer autoencoder neural network. Based on deep hidden nodes, an autoencoder can reduce features, while maintaining key data information. It then uses a scalable gradient enhancement machine to train and classify embedded features. Research focus of artificial intelligence in medicine [10], including informatics methods that range from in-depth study of information management to control of

health management systems, and active guidance of doctors in treatment decisions. The social and ethical complexity of these applications warrant continued debater to determine their utility and economic value, and to develop interdisciplinary strategies for broader use.

Looking back at these research results, it is possible to use neural network algorithms to identify breast tumors, but they fail to take into account the need for excellent training datasets for neural network models with excellent performance. In real life, neural networks are limited by sample imbalance, especially in tumor slices that are imbalanced between samples. In this way, the result of model learning will be biased towards the side of large sample size, which will lead to poor generalization ability and recognition ability of the model, which is difficult to use in clinical practice. The traditional solution to this problem is to use positive samples oversampling and negative samples in sampling. However, this results in a reduction in the amount of raw sample data. The labeling of medical sample data is time-consuming, labor-intensive and expensive, so the labelled data is very valuable. If the data volume of the original sample is reduced, resources will be wasted, so the traditional method can no longer meet the original intention of people to solve the problem.

In the identification of benign and malignant breast tumors, this paper has made the following contributions:

1. This paper predicts benign and malignant breast tumors based on convolutional neural networks, a novel computer-aided detection (CAD) system that can be used to classify benign and malignant tumors in breast tumor samples.

2. This paper uses a model based on Conditional Deep Convolutional Generative Adversarial Networks (CDCGAN), which can address the imbalance of real-life collected data by directed generation of small-sample datasets.

3. For the high-dimensional data redundancy problem, this paper proposes an integrated dimension reduction convolutional neural network (IDRCNN) model, which solves the high-dimensional data dimension reduction problem of breast cancer and extracts effective features. The subsequent classifier found that by using the IDRCNN model proposed in this paper, the recognition efficiency of the model was improved.

4. Comparing the method proposed in this paper with existing traditional methods and advanced methods, our performance is better than the current methods and is in the state-of-the-art performance.

## Methods

### Source of data

A dataset of 5547 breast histology images with a size of 50×50×3 was obtained from Andrew Janowczyk' s website and used for Epidemium data science tutorials. (http://www.andrewjanowczyk.com/use-case-6-invasive-ductal-carcinoma-idc-segmentation/).

### Data preprocessing techniques

When the training data is in an ideal state, it is easy to identify errors in some special cases, such as brightness, blur, and artifacts. Adding noise to the training data can improve the robustness of the model. This paper uses data enhancement technology. For the training pathological slice images, we use random geometric transformation methods: flip, rotate, crop, zoom, translate, and shake. This can improve the robustness of the model and reduce the sensitivity of the model to images.

## Convolutional neural network

CNN is an artificial neural network with multiple hidden layers designed by imitating the biological cerebral cortex and is comprised of a convolutional layer, pooling layer, BN layer, and a fully connected layer (Fig 1). Residual neural network 50 (ResNet50) [11] was proposed in 2015. Residual neural network refers to the idea of adding residual learning to conventional CNN, which resolves gradient dispersion and accuracy drop in the deep network. The training set problem deepens the network guaranteeing accuracy and controlling speed. Subsequently, many strategies are based on ResNet50 to complete detection, segmentation, and recognition.

## Generative adversarial network

The generative adversarial network is comprised of a generator network (generator, G) and a discriminator network (discriminator, D). The generator network generates images and randomly generates a value X. Through this value, a fake image G(X) is generated. The discriminator network identifies the image and determines if it is a generator network image or the real image. Assuming that real image input to the discriminator network is Y, the output probability value D (Y) represents whether it is a real image. The generator and discriminator belong to the process of zero-sum game in training, where the discriminator network strives to distinguish the images generated by the generator network from the real images, while the generator network strives to generate fake and real images so that the discriminator identifies network errors. Both the generator and the discriminator can be expressed by multi-layer nonlinear mapping functions using the following algorithm model. At the beginning of training, because the discriminator has a large amount of training data where D (Y) can distinguish positive vs false images, it gives the generator feedback correction information, and the modified G(X) will regenerate random values for D. The training of Y after confrontation training between the 2 to optimize loss function, is finally in a zero-sum game state. D(Y) mainly determines whether the image is from the real training set and not from the generator network, so the higher the value of D(Y), the better the model is when Y is a real image, so the smaller the value of D (G (X)), the better the model is. The generator network aims to generate a image that is as close to the real one as possible. Thus, the larger the generator network D(G(X)), the better. The formula is shown in formula (1).

$$\underset{G}{min}\ \underset{D}{max}\ V(D,G) = E_{X \sim P_{data}(X)}[logD(x)] + E_{z \sim P_Z(Z)}[log\,(1 - D\,(G\,(z)))] \tag{1}$$

Formula (1) D(G(X)) represents the probability of the generator network judging if the

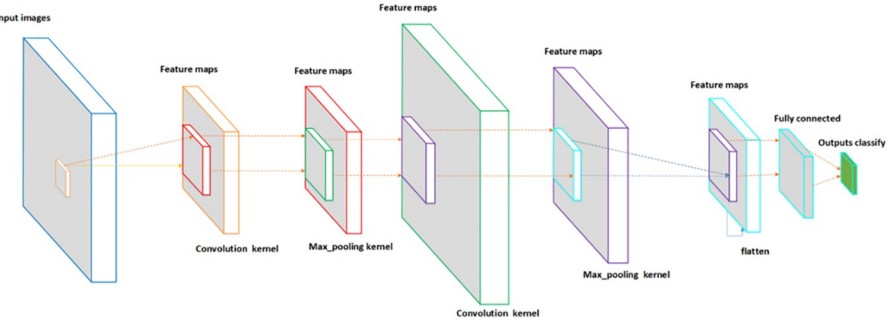

**Fig 1. The framework of CNN.**

image generated by the generator network is real. Thus, the larger the generator network, the better. Since it is a subtractive function, V (D, G) changes slightly and the result is min(G). D (Y) judges the probability that the real image is real. The higher the value of D(Y), the better the model is. Since it is an increasing function, V (D, G) becomes larger and formula (1) is max(D).

## Design of conditional deep convolution generation adversarial network model

The main method in this section uses the 2 neural networks of generation adversarial network (GAN); namely the generator network and the discriminator network. The generator network generates a small number of samples, thereby balancing sample data and, improving the model of CNN recognition rate.

However, the random value X causes the model to generate an uncertain sample. Thus, to generate a specific sample, the image and corresponding label can together, be used for training. In conditional generation adversarial network, input noise Z and label y are used as hidden layer inputs into the generator network. The formula is shown formula (2).

$$\min_{G} \max_{D} V(D, G) = E_{X \sim P_{data}(X)}[logD(x/y)] + E_{z \sim P_Z(Z)}[log\left(1 - D\left(G\left(z/y\right)\right)\right)] \tag{2}$$

In Table 1, by inputting the labeled data set (x, y), initialize x as the input training set, input x into resnet50 to extract the feature value $X_2$, input $X_2$ into the principal component analysis algorithm for dimensionality reduction to obtain $X_{PCA}$, and then $X_{PCA}$ is input into the classifier to classify benign and malignant tumors.

The output of the model is $p_\theta(X) = X_\theta$, the label is y, and the algorithm of the neural network extraction feature in IDRCNN is shown in the following formulas (3) (4) (5) (6).

$$J = \frac{1}{2}\sum_{i=1}^{m}\left(h^i_\theta(x) - y^i\right)^2 = \frac{1}{2}\left(X_\theta - y\right)\left(X_\theta - y\right)^T \tag{3}$$

$$dJ = \frac{1}{2}\mathrm{dtr}((X_\theta - y)(X_\theta - y)^T) = tr(((X_\theta - y)^T)Xd\theta) \tag{4}$$

$$\nabla_\theta J = X^T(X_\theta - y) \tag{5}$$

$$\begin{aligned} J &\to 0 \\ X_\theta = y &\to \theta = (H^TX)^{-1}X^Ty \\ loss &\to 0 \end{aligned} \tag{6}$$

**Table 1. Algorithm of integrated dimensionality reduction convolutional neural network (IDRCNN).**

| |
|---|
| **Input:** Breast cancer source data $X_s$ and their labels Y. |
| **Input:** Initialize data X as training datasets. |
| 1. X is input to the RESNET50 model to extract feature values to get $X_2$ |
| 2. Construct the $X_2$ to PCA |
| 3. Obtain the feature value $X_{PCA}$ of PCA dimension reduction. |
| 4. Put $X_{PCA}$ in the gradient boosting method. |
| 5. Get a classification result for benign and malignant tumors. |

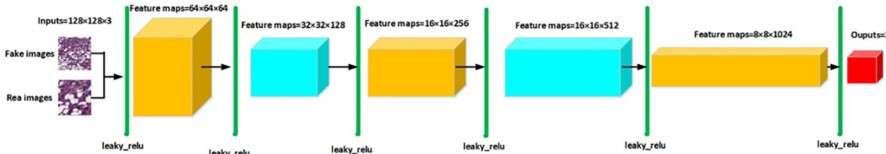

**Fig 2. The framework of discriminator model.**

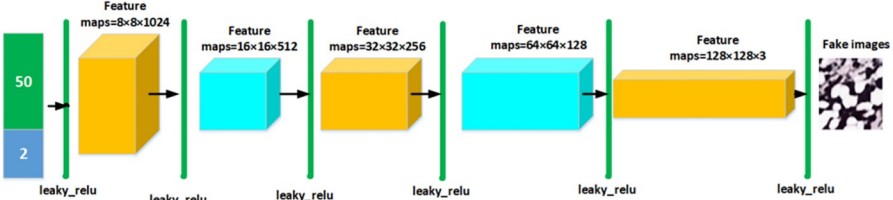

**Fig 3. The framework of generator model.**

The generator uses the input $50$ – dimensional noise and $2$ – dimensional label vector to form a $52$ –dimensional vector. The generator outputs to the $16 \times 16 \times 512$ deconvolution layer through the $8 \times 8 \times 1024$ fully connected layer, and then outputs to the BN layer. The output results in a $32 \times 32 \times 256$ deconvolution layer, and then to the BN layer and then outputs to the BN layer through $64 \times 64 \times 128$. The result is output to $128 \times 128 \times 3$. The deconvolution layer receives the generated image. The design process of the discriminator is that $128 \times 128 \times 3$ images are input into the convolutional layer $64 \times 64 \times 64$, and then output into the BN layer. Resulting in a $32 \times 32 \times 128$ convolutional layer and output to the BN layer. The obtained result is output to $16 \times 16 \times 256$ and output to the BN layer. The obtained result is output to $16 \times 16 \times 512$, and then to the BN layer. The obtained result is output to $8 \times 8 \times 1024$, and then to the BN layer. Batch normalization and leaky_relu functions are used in each deconvolution and convolution layer. The generator and discriminator images are shown in Figs 2 and 3.

## Principal components analysis

Principal components analysis (PCA) is a method of analyzing and simplifying datasets and is often used to reduce dataset dimensionality while maintaining features that contribute the most to variance in the dataset. This is done by keeping the low-order principal components and ignoring the high-order principal components.

## Gradient boosting

Gradient boosting is a machine learning technique used for regression and classification problems. It integrates weak prediction models, typically decision trees, to generate a strong prediction model. The method establishes the weak model in stages and by optimizing arbitrary differentiable loss function at each stage.

In this paper, we use random geometric transformation methods: flip, rotate, crop, zoom, translate, and shake to do data augmentation algorithm by obtaining pathological slice images as training set. Then, the CDCGAN algorithm proposed in this paper is used to generate a

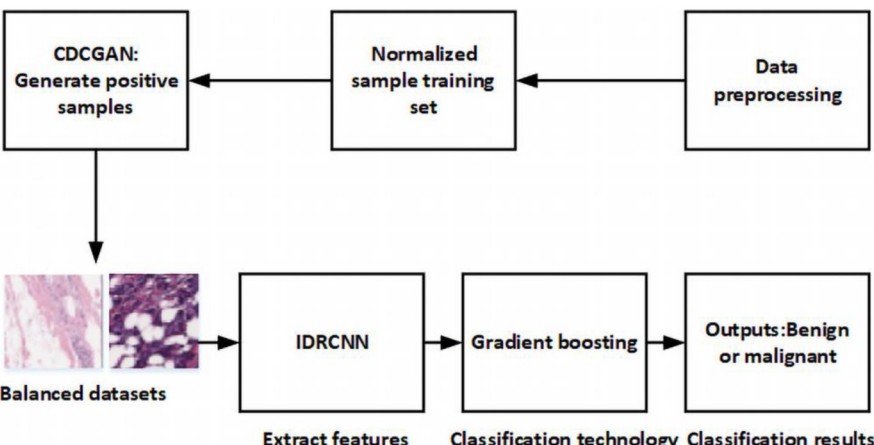

**Fig 4. The framework of our paper.**

small sample data set to obtain balanced samples. Finally, three neural network algorithms and three traditional machine learning algorithms are used to model and predict the benign and malignant results of breast tumors.The design of current study is shown in Fig 4.

## Evaluation metric

Multiple mainstream evaluation tools are available for evaluating classifiers, including confusion matrix, accuracy, receiver operating curve (ROC), area under the ROC curve (AUC), accuracy, and F1 score.

The confusion matrix is a table that shows performance classifier. Generally, in machine learning, the confusion matrix is referred to as error matrix. Depending on data type, the image area may be positive or negative. In addition, the decision on the detected result may be correct (true) or incorrect (false). Therefore, the decision will be one of four possible categories: true positive (TP), true negative (TN), false positive (FP) and false negative (FN). The correct decision is the diagonal of the confusion matrix.

$$accuracy = \frac{TP + TN}{TP + FP + TN + FN} \tag{7}$$

ROC analysis is a well-known method of evaluating detection tasks. Because of its common use in medical decision-making, ROC analysis is used in medical imaging.

The ROC curve is a graph of the operating point and can be regarded as a graph of the true false positive rate (FPR) and the true false positive rate (TPR). TPR and FPR are also called sensitivity and specificity, respectively, and are defined in formula (8) and (9).

$$sensitivity = \frac{TP}{TP + FN} \tag{8}$$

$$specificity = \frac{TN}{FP + TN} \tag{9}$$

AUC is used in medical diagnostic systems for evaluating the model based on the average value of each point on the ROC curve. For classifier performance, the AUC score should always be between "0" and "1". A model with a higher AUC value has better classifier performance.

Precision refers to the number of positive observations correctly predicted relative to the total positive observations predicted. High precision is related to low FPR. Precision is calculated as shown in formula (10).

$$specificity = \frac{TN}{FP + TN} \tag{10}$$

The F value is a weighted average of the accuracy and recall. Therefore, this value takes into account both the false positive rate and the false positive rate. The F value is defined as shown on formula (11).

$$F - value = \frac{2*Recall*Precosion}{Recall + Precision} \tag{11}$$

Positive Predictive Value (PPV) refers to the percentage of subjects who are judged to be positive (have cancer) by the early screening product. It is a measure of the ability of the early screening product to avoid "misdiagnosis" PPV = true positive population/positive test population. The ideal early screening product has a PPV of 100%, with no false positives. The PPV is defined as shown on formula (12).

$$PPV = \frac{TP}{TP + FP} \tag{12}$$

Negative Predictive Value (NPV) refers to the percentage of subjects who are judged to be negative (healthy) for early screening products and is a measure of the ability of early screening products to "miss" and exclude negatives. The NPV is defined as shown on formula (13).

$$PPV = \frac{TP}{TP + FP} \tag{13}$$

## Results

### Description of datasets

A dataset of 5547 breast histology images with a size of 50×50×3 was obtained from Andrew Janowczyk's website and used for Epidemium data science tutorials. The aim was to classify cancer images into IDC (invasive ductal carcinoma) and non-IDC images. Generally, training on many training samples enhances results quality and accuracy. However, due to the limited number of patients, the biomedical dataset contains a relatively small number of samples. Therefore, data augmentation increases input data size by generating new data from the original input data. Multiple strategies can be used for data augmentation. Here, the CDCGAN strategy was used to generate specific data images. The number of training and testing samples for each segmentation technique is shown in Table 2. All experiments were cross-validated.

Here, a breast histology image dataset was used as the training set. The input training set samples are fed into an ensemble dimensionality reduction method (IDRCNN) for

**Table 2. Summary of Breast Histology Images datasets.**

| Datasets | Training | Testing | Total |
|---|---|---|---|
| Breast Histology Images | 5492 | 555 | 5547 |
| Breast Histology Images (generate images) | 5048 | 2163 | 7211 |

dimensionality reduction of the data. The reduced dimensionality data was then imputed into gradient boosting for training and to obtain the IDRCNN model. Finally, the trained model was used to identify benign vs malignant breast tumours. The algorithm implementation process is shown in Algorithm 1. The process of categorizing the test images into normal and cancerous tissue images using the trained model. The results were validated using a 10-fold cross-check method.

## Experimental environment

The experimental platform for this paper is a server with NVIDIA GeForce RTX 1080ti GPU and 16 GB RAM, Windows system, Python 3.7, and model training is done in pycharm 64-bit operating system. The experimental model was selected from tensorflow as the learning framework, which has advantages over other frameworks in terms of flexibility, ease of use, and speed.

## Data preprocessing

The main purpose of data pre-processing is to remove pixel noise and obtain relatively less noisy image information, thus making the classification results more accurate. The main work of image data pre-processing in this paper is as follows.

1) Selection of model input image size

The training convergence time of the model is an essential indicator for evaluating the merit of the model, and three image input specifications were used to train the model.

The experimental results are shown in Table 3. The model took the best time to converge when the selected image size was 50pixel×50pixel, the features were more apparent, and the pixel distribution was more reasonable. Selection of model input image size.

2) Image preprocessing. The purpose of preprocessing is mainly to eliminate distracting factors and highlight feature information. The machine learning samples are normalized by an image preprocessing algorithm that eliminates the effect of other transform functions on the image transform using the Z-Score algorithm, whose formula is shown below.

$$\text{image}_{\text{normalization}} = \frac{\text{image} - \mu}{\sigma}$$

Data augmentation. Augmenting the image with data by rotating, panning, and scaling can effectively prevent the overfitting problem of machine learning models. In this paper, the rotation, translation and scaling operations are performed by setting the random rotation angle to [0–10 degrees], the random horizontal offset of the image to [0–0.1], the random vertical offset of the image to [0–0.1] and the random scaling of the image to [0.9,1.1]. After the above geometric transformation makes, the breast tumour image data is augmented, increasing the sample of the training set and avoiding the overfitting problem of the model due to insufficient training samples.The data-augmented images are shown in Fig 5.

3) Feature Extraction: Feature extraction of images using unique feature extraction algorithms. This paper uses the Histogram of Oriented Gradient (HOG) to extract features from

**Table 3. Different size convergence times.**

| Picture size | Convergence times/min |
|:---:|:---:|
| 224 Pixel × 224 Pixel | 900 |
| 112 Pixel × 112 Pixel | 517 |
| 50 Pixel × 50 Pixel | 194 |

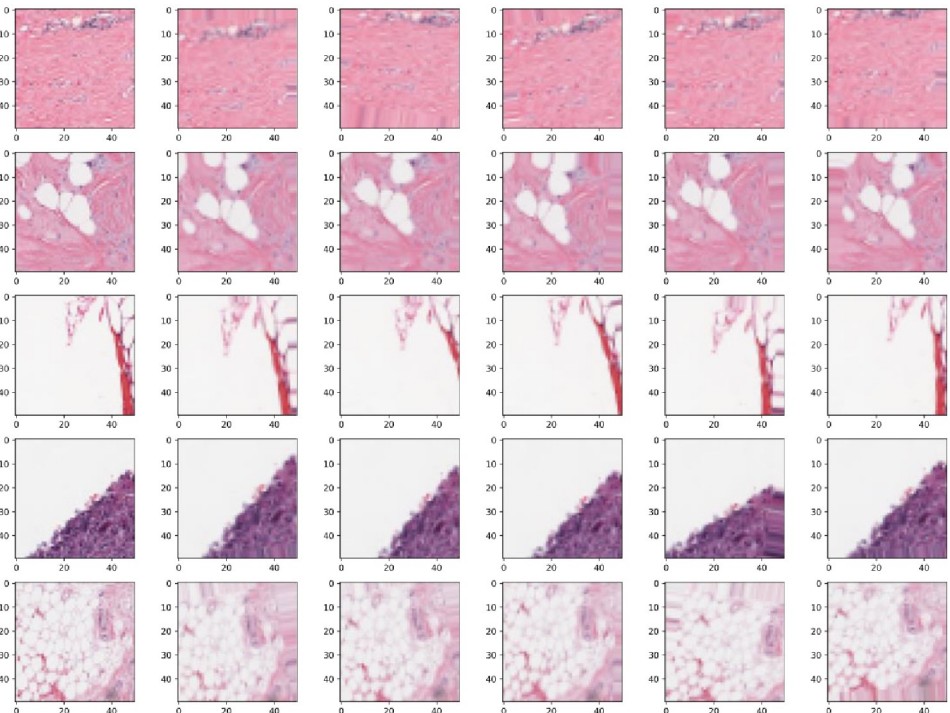

**Fig 5. Data augmentation.**

breast tumour images, which is a statistical-based feature extraction algorithm: the feature vector of an image is obtained by counting the pixels in different gradient directions. The first row shows the regularised image, and the second row shows the image after feature extraction by hog, as shown in Fig 6.

## Parameters selected

The network model is trained to achieve the optimal classification of pathological images. The model is initialized by adjusting the input format, batch size, and network learning rate. The IDRCNN model from deep learning is applied to the detection of pathological images of breast cancer. The cross-entropy loss function is selected, and the IDRCNN network structure is

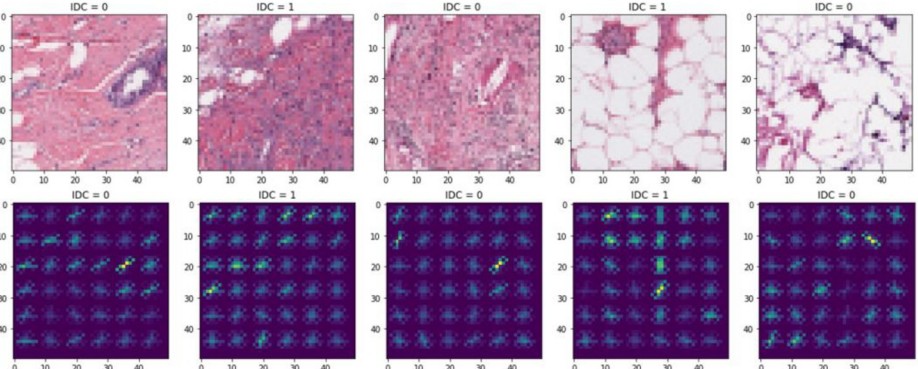

**Fig 6. Histogram of Oriented Gradient feature extraction.**

modified with some network training techniques to train a suitable model for pathological images.

1) Selection of batch_size

The selection strategy for batch size requires consideration of computer memory and GPU memory resources, the dataset's sample size, and the input images' size. For large batches of training scenarios, it is essential to ensure that more appropriate batch size is used to avoid overflowing memory. The final batch size was determined to be 100 through empirical comparison.

2) Selection of learning_rate

The learning_rate is an essential parameter in deep learning that determines whether and when the objective function converges to a local minimum. A suitable learning rate allows the target function to converge to a local minimum in a suitable amount of time. The results are shown in Fig 7. As seen from Fig 7, the loss decreases as the learning_rate decreases, i.e. the error decreases. A significant learning rate will result in skipping the optimal point and obtaining a locally optimal point, resulting in a low accuracy rate. In contrast, a small learning rate will result in a relatively low training speed but closer to the optimal point.

In order to achieve the optimal classification of pathological images, the network model was trained. The images were scaled to a uniform 50×50 and normalized into a training, validation, and test set in a ratio of 8:1:1. All experiment hyperparameters, including batch size and learning rate, were used. After several experiments with 100 training batches per training iteration, i.e. epochs of 100, it was determined that the best performance of the classification model on the validation data set was the combination of parameters with a batch size of 32 and a learning rate of 0.001, as shown in Fig 7. The learning rate update strategy used was the natural exponential decay method.

A ten-fold cross-validation method was used to validate the experimental results. The ten-fold cross-validation divides the data set into ten equal-sized parts.

Each time, a part of the data set is taken as the test set and the rest as the training set, and the average of the ten results is taken as the model's metric estimation. The principle is shown in Fig 8, and the steps of ten-fold cross-validation are

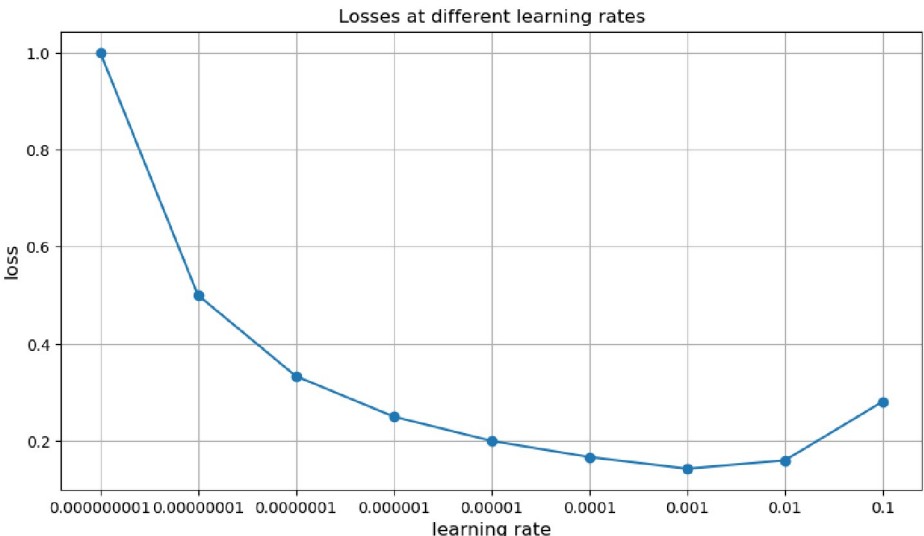

**Fig 7. Losses at different learning rates.**

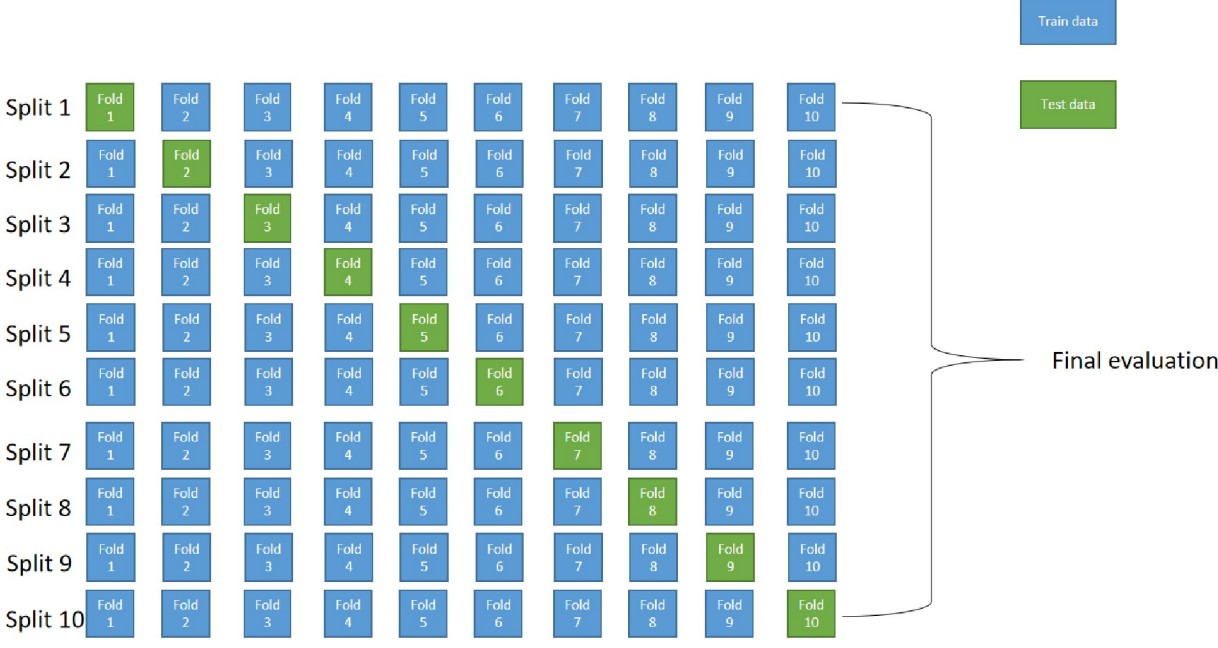

**Fig 8. Ten-fold cross-validation schematic.**

1. Divide the dataset equally into ten parts, each containing the same number of images.

2. Select one part at a time as the test set and the rest of the dataset as model input to train the model.

3. After training, record the true positives, false positives, true negatives and false negatives of the test set in turn.

4. After training, record the true positives, false positives, true negatives, and false negatives of the test set in turn.

The improved model by the proposed algorithm was compared with several other classical models, and the experimental results are shown in Table 4 and Fig 9A–9C. It can be seen that the proposed model has better AUC, sensitivity, specificity, precision,f-value,PPV,NPV and accuracy compared to other deep learning models [13–15] and machine learning models [12], indicating the feasibility of the model in the work of classifying breast cancer pathology images. After training, record the true positives, false positives, true negatives, and false negatives of the test set in turn.

**Table 4. Evaluation index parameters.**

| Methods | Accuracy (AVG) | Sensitivity (AVG) | Specificity (AVG) | Precision (AVG) | f-value (AVG) | PPV (AVG) | NPV (AVG) |
|---|---|---|---|---|---|---|---|
| CNN [15] | 0.81 | 0.67 | 0.80 | 0.68 | 0.67 | 0.77 | 0.83 |
| Machine learning [12] | 0.75 | 0.65 | 0.73 | 0.65 | 0.64 | 0.76 | 0.77 |
| Resnet50 [14] | 0.83 | 0.81 | 0.81 | 0.75 | 0.79 | 0.81 | 0.79 |
| AlexNet [13] | 0.80 | 0.80 | 0.75 | 0.78 | 0.82 | 0.81 | 0.81 |
| Proposed model | **0.85** | **0.81** | **0.83** | **0.82** | **0.84** | **0.83** | **0.82** |

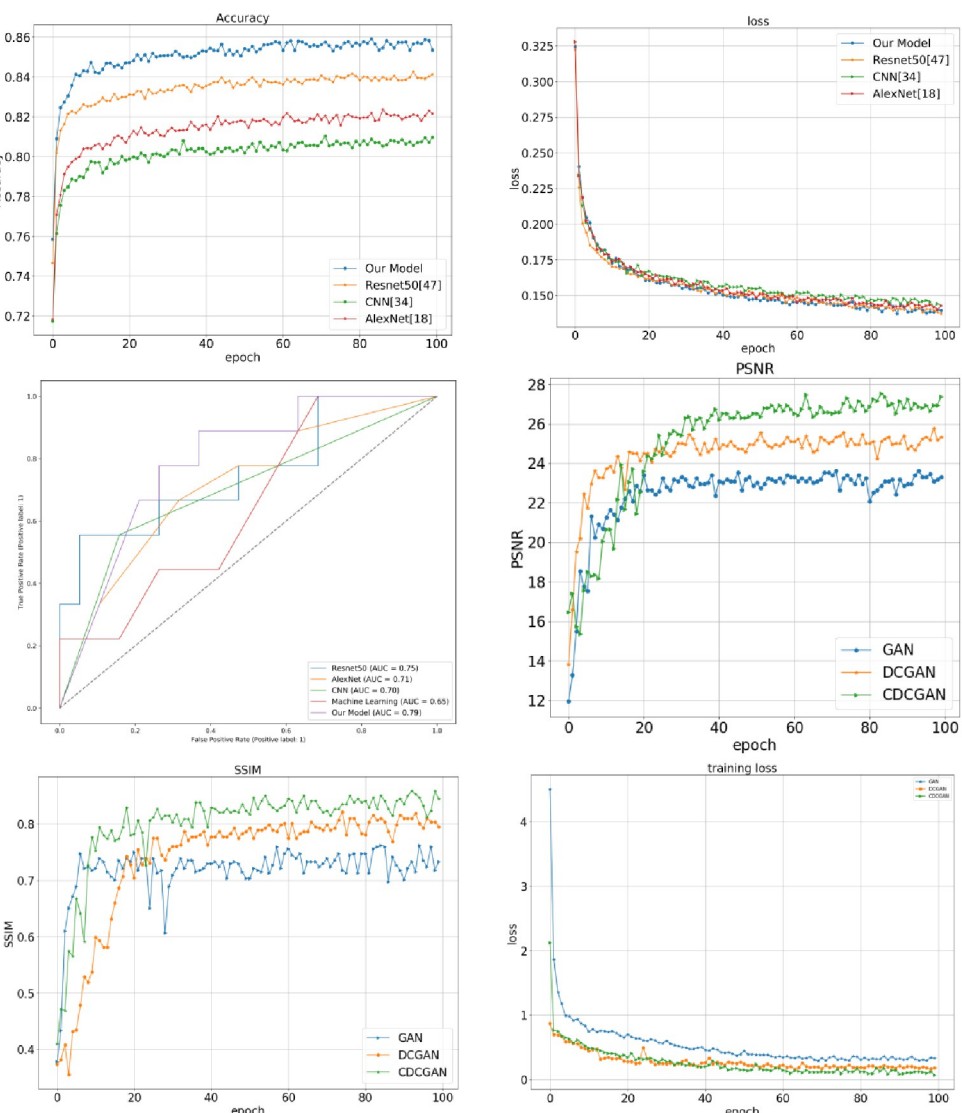

**Fig 9. The performance of training dataset and test dataset.** (A) The accuracy of training dataset. (B) The loss of training dataset. (C) The AUC of test dataset. (D) The peak signal-to-noise ratio (PSNR) of Training dataset. (E)The structural similarity (SSIM) of Training dataset. (F) The loss of Training dataset.

## Similarity check between real and fake images

This study used PSNR (peak signal-to-noise ratio), SSIM (structural similarity), and loss function to evaluate the quality of the generated samples, with larger PSNR values indicating better quality samples and larger SSIM values indicating better visual results. Table 5 shows the PSNR and SSIM evaluation values.

As can be seen from Table 5 and Figs 9D–9F and 10, the GAN loss function converged steadily at 100 iterations. The generated breast tumour slice samples were ambiguous because the GAN [16] generative adversarial network training did not thoroughly learn the breast tumour sample features, with a PSNR value of 22.36 dB and SSIM of 0.7632. DCGAN [17] converged faster than the GAN model, indicating that using a deep convolutional neural network accelerated the model training speed and improved the model feature extraction

**Table 5. The PSNR and SSIM evaluation values.**

| Method | PSNR (dB) | SSIM |
|---|---|---|
| GAN [16] | 22.36 | 0.7632 |
| DCGAN [17] | 24.44 | 0.8033 |
| CDCGAN | 27.35 | 0.8422 |

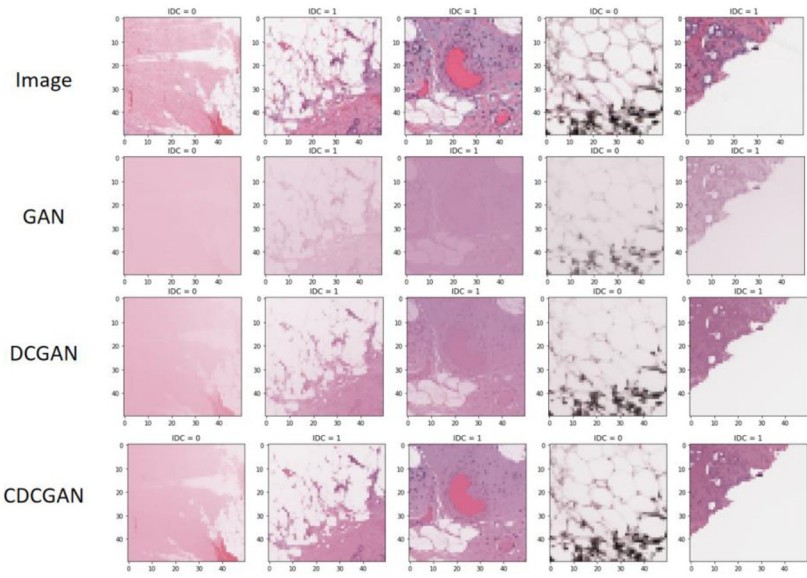

**Fig 10. The result of generate images in different models.**

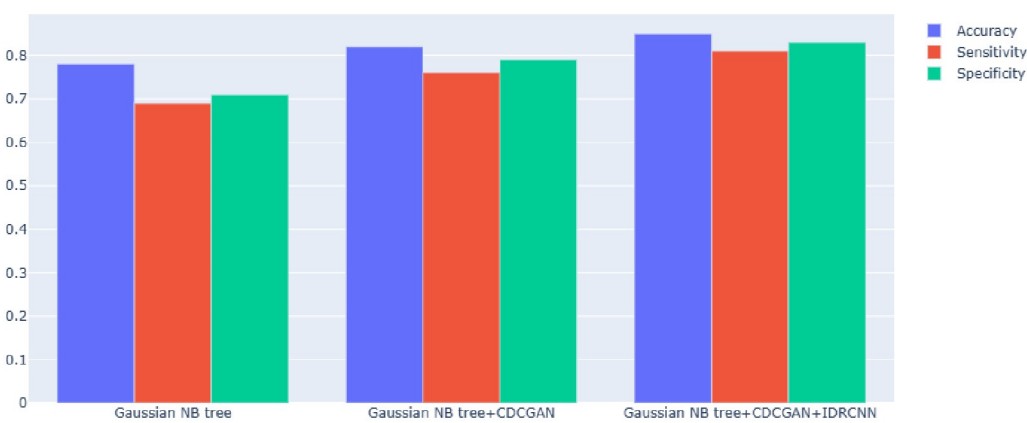

**Fig 11. The results of the ablation experiment.**

capability. The faster convergence speed of DCGAN compared to the GAN model indicated that the use of a deep convolutional neural network accelerated the training speed of the model and improved the feature extraction ability of the model, which further improved the ability of DCGAN to generate samples with PSNR value of 24.44 dB and SSIM value of 0.8033. The generated breast tumour samples had more apparent target edges and better visual effects with higher quality. The CDCGAN adds a conditional mechanism to the DCGAN to further

focus on the practical breast tumour sample features, resulting in more precise details of the generated breast tumour texture with a PSNR of 27.35 dB and SSIM of 0.8422. This further illustrates the effectiveness of the CDCGAN conditional deep convolutional neural network proposed in this study.

## Ablation experiments

In the experiment part, ablation experiments were conducted on the data set of Breath Histology Images datasets for IDRCNN, CDCGAN, and Gaussian NB trees, and the following models were constructed for ablation experiments: (1) Gaussian NB trees were used to classify breast tumour images; (2) Gaussian NB tree combined with CDCGAN for breast tumour image classification; (3) Gaussian NB tree combined with CDCGAN and IDRCNN to classify breast tumour images; In order to ensure the fairness of the comparison experiment, the equipment and parameter settings used in the three groups of experiments are the same. The experimental results are shown in Fig 11 and Table 6.

The Gaussian NB tree represents the case without the participation of CDCGAN and IDRCNN, and the accuracy decreases by 7% compared to the proposed method. The Gaussian NB tree + CDCGAN decreases by 3% compared to the proposed method. CDCGAN is used to increase the number of training samples and thus improve the robustness of the model. In contrast, IDRCNN is used to reduce the dimensionality of the data and extract compelling features, thus providing the accuracy of the model and thus improving the performance of breast tumour image classification.

Using the IDRCNN method proposed in this paper to randomly generate predicted results and accurate result image comparison reveals that the proposed method has high accuracy. The comparison chart of the predicted value and the actual value is shown in Fig 12. It shows that the method proposed in this paper can predict the results more accurately and can be used as an automatic clinical diagnosis.

## Discussion

We constructed AI models to classify breast tumours, and the IDRCNN model showed high performance. Early breast cancer detection and treatment are critical for good outcomes. The breast tumour detection approach proposed here can accurately determine the patient's condition early. This approach solves the medical imbalance and minor sample feature caused by differences in breast cancer incidence, which makes it challenging to train an effective and accurate breast cancer prediction model. In this paper, the problem of unbalanced data can be solved by generating small sample image data via deep convolution to generate an adversarial network. The IDRCNN method solves the complex problem of model training caused by high-dimensional feature values and finally connects the gradient boosting method for classification. Experimental comparison of the proposed method with the methods proposed in other studies showed that the strategy proposed in this article performs well in terms of f-value,accuracy,AUC,sensitivity,specificity, precision,PPV,NPV and AUC.

**Table 6. Ablation experiments.**

| Gaussian NB tree | CDCGAN | IDRCNN | Accuracy↑ | Sensitivity↑ | Specificity↑ |
|:---:|:---:|:---:|:---:|:---:|:---:|
| | | | (AVG) | (AVG) | (AVG) |
| √ | | | 0.78 | 0.69 | 0.71 |
| √ | √ | | 0.82 | 0.76 | 0.79 |
| √ | √ | √ | **0.85** | **0.81** | **0.83** |

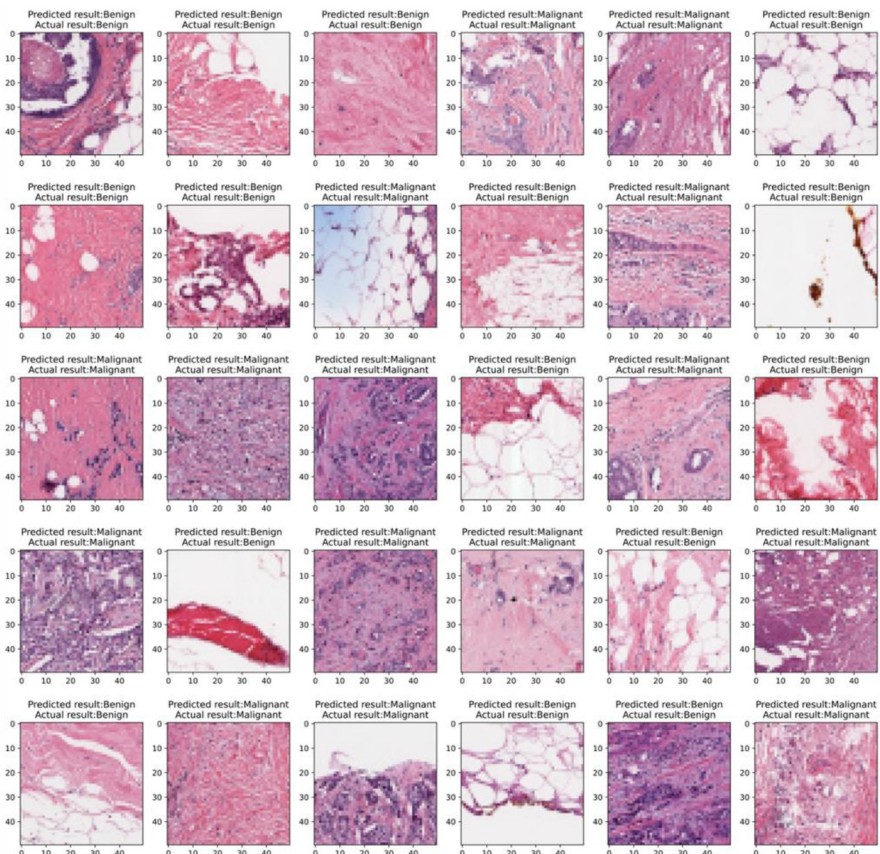

**Fig 12. The results of predicted value and real value.**

Early tumor detection remains a challenging to timely cancer diagnosis. Tumors may be benign or malignant. Early tumors detection is crucial for successful cancer treatment. Cancer diagnosis typically involves microscopic analysis of tumor sections by qualified pathologists. Jian-Xing WU [18] integrated 2D Harris corner convolution, maximum buffering and random decision forest into machine vision classifiers to identify benign or malignant tumors. Ziqi Yang [19] developed a technique called temporary sequential dual branch network that uses both B-mode ultrasound data and contrast-enhanced ultrasound data. Hilal M. El Misilmani [20] improved the resolution and accuracy of magnetic detection electrical impedance tomography and develop an effective imaging technique for breast cancer diagnosis.

Despite the proliferation of new technologies, the high computation and sample volumes posed significant challenges. Ruijuan Chen [21] proposed a new algorithm based on stacked auto encoder neural network. The disadvantage of this approach is that the experimental environment requires high computation and is difficult to conduct. The study by Roslidar Roslidar [22] is based on progress in breast cancer detection using thermal imaging and CNN. Abdelhameed Ibrahim [7] proposed a thermal imaging breast cancer segmentation method based on Chaos bee colony algorithm. Although the Salp Swarm algorithm has advantages in single-objective optimization problems, its convergence speed is low, stagnating local optimum. Nan Wu [23] used a ResNet-based custom network as building block. Depth and width balance was optimized for high-resolution medical images. The advantage of this strategy is that the residual network is proposed as the model's building block. Meriem Sebai [24] developed a method that uses a partially supervised deep learning framework to automatically detect mitotic maps

in breast cancer histological sections. Gege Ma [25] introduced and evaluated spectral capacitive coupled resistivity tomography for breast cancer. However, this approach has higher tools requirements. An expert system based on artificial neural networks, could detect bundle branch block disease using the above method [26]. Halefom Tekle Weldegebriel [27] proposed a hybrid model of 2 super classifiers: CNN and extreme gradient boosting (XGBoost) where CNN acts as a trainable automatic feature extractor of the original image. XGBoost uses the extracted features as input for recognition and classification. Relative to traditional fully connected layers, XGBoost can generate better results. Ghazanfar Latif [28] proposed a CNN to despeckle ultrasound images, and proposed another CNN model for classifying ultrasound images into benign and malignant categories. Chuang Zhu [29] proposed breast cancer histopathological image classification assembling multiple compact CNN. A novel method that uses deep learning and segmentation techniques to classify benign vs malignant breast cancer masses on mammograms has been proposed [30, 31]. The principle uses NN as a feature extractor, and then support vector machine to classify results [32].

In conclusion, we developed a breast cancer diagnosis model that utilized generative adversarial networks to address data imbalance. The positive samples were generated by the conditional generative adversarial network to obtain a balanced data, then the feature values were extracted by IDRCNN method. Finally, a classification technique gradient boosting algorithm was used to distinguish benign and malignant tumors. These processes improve the performance and generalization ability of the classifier. Finally, our proposed method achieves 85% accuracy in the Breast Histology Images dataset. Compared with traditional convolutional neural networks, the IDRCNN model achieves 4% higher accuracy on the Breast Histology Images dataset. Our proposed method has reference value for designing structural diversity using other types of basic learners such as decision trees, residual networks, convolutional neural networks. The method can also be used for the detection of other types of cancer, providing early diagnosis guidance for doctors, and has many useful medical applications in clinical. For patients with a history of breast cancer, this model could bring faster intervention and the most appropriate treatment.

The study had some limitations. Firstly, it was only conducted with a single data set and small sample. Secondly, the model lacked interpretability and parameters. In future, we plan to validate on multiple other pathological datasets, improve the generalization ability of the model, and consider enhancing the interpretability of the model, collect data on multiple centers to validate the model, and improve the robustness of the model.

## Conclusion

In this study, we proposed a CDCGAN which can solve the imbalance problem of manually collected data by directionally generating small sample data sets. Experimental results show that IDRCNN combined with the model of CDCGAN model has superior classification performance than existing methods, as revealed by sensitivity, area under the curve (AUC), ROC curve and accuracy, recall and f-value analysis.

## Author Contributions

**Conceptualization:** HuanQing Xu.

**Data curation:** HuanQing Xu.

**Formal analysis:** HuanQing Xu, Xian Shao.

**Funding acquisition:** Li Jin.

**Investigation:** Xian Shao.

**Methodology:** Li Jin.

**Project administration:** Li Jin.

**Resources:** Li Jin.

**Software:** HuanQing Xu, Li Jin.

**Supervision:** Li Jin.

**Validation:** Li Jin.

**Visualization:** Li Jin.

**Writing – original draft:** HuanQing Xu, Li Jin.

**Writing – review & editing:** Xian Shao, Shiji Hui.

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
