## [Decision Letter · Decision Letter 0]

6 Nov 2022

PONE-D-22-27228Supervised Breast Cancer Prediction using Integrated Dimensionality Reduction Convolutional Neural NetworkPLOS ONE

Dear Dr. Jin,

Thank you for submitting your manuscript to PLOS ONE. After careful consideration, we feel that it has merit but does not fully meet PLOS ONE’s publication criteria as it currently stands. Therefore, we invite you to submit a revised version of the manuscript that addresses the points raised during the review process.

We look forward to receiving your revised manuscript.

Kind regards,

Vijayalakshmi G V Mahesh, Ph.D

Academic Editor

PLOS ONE

Journal Requirements:

2. Please note that PLOS ONE has specific guidelines on code sharing for submissions in which author-generated code underpins the findings in the manuscript. In these cases, all author-generated code must be made available without restrictions upon publication of the work. 

Please review our guidelines at https://journals.plos.org/plosone/s/materials-and-software-sharing#loc-sharing-code and ensure that your code is shared in a way that follows best practice and facilitates reproducibility and reuse. New software must comply with the Open Source Definition.

"The research is supported by the National Natural Science Foundation of China (No. 61602005, 61572030), 973 Project (No. 2015CB351705), Natural Science Foundation of Anhui Province (1608085MF130, 1808085MF199), Humanities and Social Sciences in Universities of Anhui Province (KJ2018A0016), Anhui University Doctor Startup Fund."

Please state what role the funders took in the study.  If the funders had no role, please state: ""The funders had no role in study design, data collection and analysis, decision to publish, or preparation of the manuscript."" If this statement is not correct you must amend it as needed. 

6. Please amend either the abstract on the online submission form (via Edit Submission) or the abstract in the manuscript so that they are identical.

Reviewers' comments:

Reviewer's Responses to Questions

**Comments to the Author**

1. Is the manuscript technically sound, and do the data support the conclusions?

Reviewer #1: Yes

Reviewer #2: Partly

2. Has the statistical analysis been performed appropriately and rigorously? 

Reviewer #1: No

Reviewer #2: Yes

3. Have the authors made all data underlying the findings in their manuscript fully available?

Reviewer #1: Yes

Reviewer #2: Yes

4. Is the manuscript presented in an intelligible fashion and written in standard English?

Reviewer #1: Yes

Reviewer #2: Yes

5. Review Comments to the Author

Reviewer #1: 1.There is limitation on statistical analysis of evaluation metrics from previous works

2.The similarity check between the real and fake images are not presented

3.Since the paper has work related on traditional machine learning, the features,the training samples and test samples considered for traditional machine learning are not presented.

4.The papers has not mentioned on any other training options such as learning rate

Reviewer #2: 1. Authors need to demonstrate the robustness of their algorithm in comparison with existing algorithms. This can be done in the results section. It may be noted that, in Table 3 Evaluation Index Parameters, the results of proposed algorithm are mentioned. How these results are superior to findings of other researchers. Authors need to refer the similar works in terms of their results.

2. In the data preprocessing section, authors mentioned general geometric transformations. It is required to explain the specific details of transformations used.

3. For validation of the results, does authors used a totally new dataset other than training and testing data? or any cross validation techniques are used? provide specific details on validation methods adopted.

6. PLOS authors have the option to publish the peer review history of their article (what does this mean?). If published, this will include your full peer review and any attached files.

Reviewer #1: No

Reviewer #2: No

---

## [Author Response · Author response to Decision Letter 0]

11 Jan 2023

Dear editor and reviewers,

We would like to express our sincere gratitude to you for your efforts in handling our manuscript. We appreciate you give us the opportunity to revise our paper. We thank the editorial board’s and the two reviewer’s points and suggestions to improve our work. Those comments are all valuable and very helpful for revising and improving our paper. 

We have studied comments carefully and have made correction which we hope meet with approval. Based on your comment and request, we attached revised manuscript with the correction sections red marked. We also replied the reviewers’ questions point-by-point as following below. The corrections in the paper and the responds to the reviewer’s comments are as following. 

For further details, please see the response letter, which is attached to the bottom of the manuscript.

Should you have any questions, please contact us without hesitate.

Once again, thank you very much for your constructive comments and suggestions which would help us both in English and in depth to improve the quality of the paper.

Yours sincerely,

Huanqing Xu

2022/12/21

---

## [Decision Letter · Decision Letter 1]

14 Feb 2023

Supervised Breast Cancer Prediction using Integrated Dimensionality Reduction Convolutional Neural Network

PONE-D-22-27228R1

Dear Dr. Jin,

We’re pleased to inform you that your manuscript has been judged scientifically suitable for publication and will be formally accepted for publication once it meets all outstanding technical requirements.

Kind regards,

Vijayalakshmi G V Mahesh, Ph.D

Academic Editor

PLOS ONE

Reviewers' comments:

Reviewer's Responses to Questions

**Comments to the Author**

1. If the authors have adequately addressed your comments raised in a previous round of review and you feel that this manuscript is now acceptable for publication, you may indicate that here to bypass the “Comments to the Author” section, enter your conflict of interest statement in the “Confidential to Editor” section, and submit your "Accept" recommendation.

Reviewer #1: All comments have been addressed

Reviewer #2: All comments have been addressed

2. Is the manuscript technically sound, and do the data support the conclusions?

Reviewer #1: Yes

Reviewer #2: Yes

3. Has the statistical analysis been performed appropriately and rigorously? 

Reviewer #1: Yes

Reviewer #2: Yes

4. Have the authors made all data underlying the findings in their manuscript fully available?

Reviewer #1: Yes

Reviewer #2: Yes

5. Is the manuscript presented in an intelligible fashion and written in standard English?

Reviewer #1: Yes

Reviewer #2: Yes

6. Review Comments to the Author

Reviewer #1: All supporting details for all the reviewer comments has been revised and addressed by the authors

Reviewer #2: All the suggestions that I made in my earlier review is answered and incorporated in the current version.

7. PLOS authors have the option to publish the peer review history of their article (what does this mean?). If published, this will include your full peer review and any attached files.

Reviewer #1: No

Reviewer #2: **Yes: **Saneesh Cleatus T

---

## [Editor Report · Acceptance letter]

26 Apr 2023

PONE-D-22-27228R1 

Supervised Breast Cancer Prediction using Integrated Dimensionality Reduction Convolutional Neural Network 

Dear Dr. Jin:

I'm pleased to inform you that your manuscript has been deemed suitable for publication in PLOS ONE. Congratulations! Your manuscript is now with our production department. 

Kind regards, 

on behalf of

Dr. Vijayalakshmi G V Mahesh 

Academic Editor

PLOS ONE